# Mental Health Symptom Reduction Using Digital Therapeutics Care Informed by Genomic SNPs and Gut Microbiome Signatures

**DOI:** 10.3390/jpm12081237

**Published:** 2022-07-28

**Authors:** Inti Pedroso, Shreyas Vivek Kumbhare, Bharat Joshi, Santosh K. Saravanan, Dattatray Suresh Mongad, Simitha Singh-Rambiritch, Tejaswini Uday, Karthik Marimuthu Muthukumar, Carmel Irudayanathan, Chandana Reddy-Sinha, Parambir S. Dulai, Ranjan Sinha, Daniel Eduardo Almonacid

**Affiliations:** 1Digbi Health, Mountain View, CA 94040, USA; inti@digbihealth.com (I.P.); shreyas@digbihealth.com (S.V.K.); bharat@digbihealth.com (B.J.); santosh@digbihealth.com (S.K.S.); ssingh.rambiritch@gmail.com (S.S.-R.); tejaswiniud@gmail.com (T.U.); karthik@digbihealth.com (K.M.M.); carmel@digbihealth.com (C.I.); chandana@digbihealth.com (C.R.-S.); ranjan@digbihealth.com (R.S.); 2National Centre for Cell Science, Pune 411007, India; dattamongad@gmail.com; 3Division of Gastroenterology, Northwestern University, Chicago, IL 60208, USA; parambir.s.dulai@gmail.com

**Keywords:** anxiety, depression, insomnia, gut-brain-axis, multi-omic models, non-pharmacological treatment

## Abstract

Neuropsychiatric diseases and obesity are major components of morbidity and health care costs, with genetic, lifestyle, and gut microbiome factors linked to their etiology. Dietary and weight-loss interventions can help improve mental health, but there is conflicting evidence regarding their efficacy; and moreover, there is substantial interindividual heterogeneity that needs to be understood. We aimed to identify genetic and gut microbiome factors that explain interindividual differences in mental health improvement after a dietary and lifestyle intervention for weight loss. We recruited 369 individuals participating in Digbi Health’s personalized digital therapeutics care program and evaluated the association of 23 genetic scores, the abundance of 178 gut microbial genera, and 42 bacterial pathways with mental health. We studied the presence/absence of anxiety or depression, or sleep problems at baseline and improvement on anxiety, depression, and insomnia after losing at least 2% body weight. Participants lost on average 5.4% body weight and >95% reported improving mental health symptom intensity. There were statistically significant correlations between: (a) genetic scores with anxiety or depression at baseline, gut microbial functions with sleep problems at baseline, and (b) genetic scores and gut microbial taxa and functions with anxiety, depression, and insomnia improvement. Our results are concordant with previous findings, including the association between anxiety or depression at baseline with genetic scores for alcohol use disorder and major depressive disorder. As well, our results uncovered new associations in line with previous epidemiological literature. As evident from previous literature, we also observed associations of gut microbial signatures with mental health including short-chain fatty acids and bacterial neurotoxic metabolites specifically with depression. Our results also show that microbiome and genetic factors explain self-reported mental health status and improvement better than demographic variables independently. The genetic and microbiome factors identified in this study provide the basis for designing and personalizing dietary interventions to improve mental health.

## 1. Introduction

Poor mental health is a significant determinant of health-related quality of life with implications at individual and population levels. Pharmacological and behavioral interventions focused on prevention or treatment of individuals suffering from mental health disorders have limited efficacy, and many experience relapse [1]. The COVID-19 pandemic resurfaced and worsened the mental health crisis and brought awareness to society of its relationship with obesity and other chronic health conditions [2]. There is a great need to develop cost-effective interventions that provide significant short- and long-term therapeutic benefits to individuals suffering from mental health, especially those with multiple comorbid conditions. Digital therapeutics have gained increased attention as a strategy to provide care to large numbers of individuals, and emerging evidence suggests its effectiveness for several chronic diseases [3,4].

There is compelling evidence indicating that genetic and non-genetic factors contribute to the etiology of mental health disorders [5,6,7], and also substantial evidence supports the link between mental health with obesity, digestive, and gut disorders [8,9,10]. For instance, two meta-analyses found a higher rate of anxiety and depression among IBS patients [11,12], and a recent genetic study identified genetic factors linking IBS with mental health disorders [13]. Another meta-analysis found a bidirectional relationship between depression and obesity, with depressed individuals having a 37% increased risk of being obese and obese individuals having an 18% increased risk of being depressed [14].

Recently, extensive epidemiological studies coupled with gut microbiome sequencing are reinforcing the importance of the gut–brain axis and identifying the gut microbiome factors underlying it [15]. A gut microbiome and behavioral study found associations between the gut bacterial taxa and their metabolic functions with depression and quality of life [16]. The current evidence shows that interindividual variation in the gut microbiome composition is mainly due to non-genetic factors, i.e., diet, exercise, and medication [17,18,19] and, therefore, the gut microbiome primarily contributes to the non-genetic etiology of mental health, obesity, and digestive disorders. Thus, modulation of the gut microbiome by dietary and lifestyle interventions is a promising avenue for preventing and treating mental health (for instance, see [20,21,22,23,24]); however, there is no clear consensus around their efficacy [25,26,27,28,29,30]. It is plausible that interindividual differences, mediated by genetic or non-genetic factors such as the gut microbiome, can explain why some individuals improve their mental health after weight loss and others do not.

Digbi Health has implemented a personalized digital care intervention that uses its multi-omics platform to provide dietary and lifestyle recommendations personalized using genetic and microbiome information. The intervention has been shown to deliver body weight loss in over 70% of individuals [31], reduction of fasting blood glucose level by 17.55% and an average reduction of HbA1c level by 6.27% [32], and a significant reduction in the symptomatology of functional gastro-intestinal disorders (FGIDs) [33]. This study addresses if gut microbiome taxa or functions and genetic markers explain body weight loss’s effect on mental health. Our results provide evidence for genetics and gut microbiome factors that drive improvement in mental health after weight loss. They offer an opportunity to personalize and develop tailored dietary recommendations to tackle obesity and mental health disorders.

## 2. Materials and Methods

### 2.1. Participant Enrollment, Intervention, and Phenotype Data Collection

Study participants were recruited from February 2020 to October 2021 among those who achieved 2% or more body weight loss from the date when enrolled in the Digbi Health personalized digital care program. Participants filled out an online questionnaire regarding anxiety or depression and sleep problems at baseline. Those who indicated a positive answer to having either of the conditions, answered questions regarding symptom intensity at baseline (T0) and follow-up (T1) on a scale from 0 (minimum) to 5 (maximum). Additionally, at baseline, participants provided information on the presence or absence of symptoms associated with FGIDs, prescribed and over-the-counter medications or supplements, alcohol intake, recreational drug usage, and demographic information, including age, gender, and height. Self-reported medication was used to identify those taking antidepressants, antianxiolytics, antibiotics, or antimycotics (see Appendix A).

Digbi Health’s intervention has been described elsewhere [33]. In a nutshell, personalization of dietary plans is achieved by analyzing participants’ genetics, gut bacteria, lifestyle, and demographics. Based on these data, the program encourages participants to make incremental lifestyle changes focused on reducing sugar consumption and timing meals to optimize insulin sensitivity, reduce systemic inflammation by identifying possibly inflammatory and anti-inflammatory nutrients, and increase fiber diversity to improve gut health. Behavioral changes are implemented with the help of virtual health coaching and the app, ensuring they are habit-forming.

Subjects self-collected saliva samples using buccal swabs and fecal samples using fecal swabs using standard methods. DNA genotyping and genotype calling, and processing of baseline (pre-intervention) fecal samples by 16S rRNA gene amplicon sequencing were performed at Akesogen Laboratories in Atlanta, GA, USA.

### 2.2. Data Analyses

The bacterial 16S rRNA gene V3-V4 region was amplified and sequenced on the Illumina MiSeq platform using 2 × 300 bp paired-end sequencing and sequence reads were demultiplexed, and ASVs were generated using DADA2 in QIIME2 version 2020.8 [34]. Quality control steps included removal of primers and low-quality bases, removal of hits to non-bacterial sequences, any ASVs not seen more than once in 10% of samples, and unassigned taxa at the phylum level [35]. Six samples were excluded from the downstream analysis due to participants reporting antibiotic consumption. The abundance matrix was rarefied at even depth (n = 36,000 reads per sample, using 500 iterations) using QIIME2 [36] and abundances were agglomerated at the genus level, resulting in 178 taxa across 344 samples. The abundance of microbial functional pathways related to neuroactive metabolites [16] was calculated with the q2-picrust2 plugin (version 2021.2) in QIIME2 [37] and the Omixer-RPM package (version 0.3.2) [38]. All raw abundances were centered-log ratio (CLR) transformed [39].

Probe level DNA genotype call files were formatted in VCF format with QC steps including removal of discordant genotypes and left normalization. Beagle version 5.3 [40,41] was used for phasing and imputation using the 1 KG project as reference panel [42] resulting in 13,478,023, chip-genotyped and those variants with imputation r^2^ ≥ 0.8, that were used on downstream analyses. For population structure analyses, we calculated the first 20 principal components and estimated genetic ancestry components using the 1 KG project samples (see Appendix A). We calculated genetic scores for 23 traits selected because of their comorbidity with obesity, digestive system (IBS and IBD), and mental health disorders (anxiety and sleep) [43,44] (see Appendix A). All genetic scores were coded to be interpreted such that a larger genetic score is associated with increasing inherited genetic predisposition to the condition.

We used logistic regression for binary outcomes and Poisson regression to model improvement of symptom intensity scores, with the outcome being the intensity at T1 and offset being the intensity at T0 with the HC3 covariance matrix as recommended elsewhere [45]. In all models, we included as demographic variables FGID: binary, the self-reported status of the previous diagnosis of a functional gastrointestinal disorder; gender: binary, male or female; BMI at T0: continuous variable; age: continuous variable; weight loss: categorical variable, categorized as those with no change, lost 0 to 5%, 5–10%, or more than 10% of their body weight at T1 in relation to T0; and five principal components (continuous variable) calculated using the genetic ancestry analyses described previously. On the logistic models, a regression coefficient greater than zero is interpreted as an increasing prevalence of self-reported illness with an increasing abundance of microbiome factors or a higher value of the genetic scores. On the Poisson regression models, a regression coefficient greater than zero is interpreted as a higher abundance of microbiome factors or a higher value of genetic scores being associated with less than average improvement in the outcome. We selected statistically significant results using an FDR ≤ 0.15 [46]. By performing multivariate analysis using PERMANOVA [47] and linear regression models, we tested the effect of potential confounders (medications and alcohol consumption) on the association of gut microbiome with mental health outcomes. We compared the relative ability of demographic (D), microbiome (M), and genetic (G) predictors to explain baseline and improvement on mental health outcomes by building models including only D, D + M, D + G, and D + M + G variables. These four models were compared using Cox-Snell pseudo r-squared values corrected by the number of predictors using Pratt’s method [48] with variability of the pseudo r-squared estimated using bootstrap. Appendix A provide additional information on each of the methods used in this study.

## 3. Results

### 3.1. Data Collection

The study sample consisted of 369 individuals recruited from the Digbi Health research study cohort who self-reported anxiety/depression or sleep problems when starting Digbi Health personalized digital care intervention (Appendix A). The study subjects had been on the intervention on average 88.3 days (median = 64 and std = 67.7 days) by the time they lost at least 2% body weight. A health questionnaire was provided and answered on average 1.7 days (median = 0 and std = 7.7 days) after receiving it. We obtained microbiome and genetic data for 344 and 348 individuals, respectively, and 328 submitted both sample types. We included in the analyses 178 bacterial genera and 42 functional pathways from the gut microbiome samples, and 23 genetic scores and ancestry from the genetic samples. For the baseline models of anxiety or depression and sleep problems, all these 328 participants were studied. For the improvement models, we started with the 147, 148, and 163 individuals that reported their change in intensity for anxiety, depression, and insomnia, respectively. Most individuals (>95%) reported improvement or maintenance of symptoms. Only 4% (6 out of 148) for depression, 2.7% (4 out of 147) for anxiety, and 1.8% (3 out of 163) for insomnia reported higher scores at T1 compared with T0 (Appendix A). Due to the small sample size of individuals with worsening symptoms, we excluded them from additional analyses. Likewise, we also excluded individuals reporting improvements in 4 and 5 scale points, which were 8 for anxiety, 7 for depression, and 6 for insomnia (Appendix A). Thus, improvement models were based on 135 responses for anxiety, 135 for depression, and 154 for insomnia.

### 3.2. Cohort Demographic Characteristics

Appendix A summarizes the demographic variables used in the analyses performed in the study. The mean BMI of the participants was 34.6, corresponding to obese class 1 individuals and in agreement with the fact that most Digbi Health care digital program participants undergo a dietary intervention to lose weight. When answering the follow-up questionnaire, most of the participants (60.4%) lost between 5 to 10% body weight. This cohort had a high prevalence of individuals with FGIDs (84%) and females (79%). A total of 284 (86%) of the participants reported taking antidepressants (40 or 12%) or antianxiolytics (244 or 74%) at baseline. Appendix A provides demographic information stratified by the level of improvement in anxiety, depression, or insomnia at T1 compared with T0. Genetic ancestry analyses identified the majority of individuals as of European ancestry (43%), followed by Africans (28%), Americans (20%), East Asians (7%), and Southeast Asians (1%). We included the first five principal components from the genetic ancestry analysis as covariates in all analyses.

### 3.3. Baseline Gut Microbiome and Genetic Factors Are Associated with Mental Health Improvement after Dietary Intervention

After the dietary and lifestyle intervention, 59% of the 135 individuals studied reported improving their anxiety symptoms, with 22%, 24%, and 13% reporting improvement in 1, 2, and 3 scale points, respectively, and 41% reporting no improvement. Three genetic scores, irritable bowel syndrome (IBS), body mass index (BMI), and obstructive sleep apnea (OSA), were noted to be significantly associated with changes in anxiety (Table 1). IBS correlated with less than average improvement and BMI and OSA with greater than average improvement in anxiety intensity scores after intervention. Seven bacterial genera were statistically associated with changes in anxiety. The abundance of four, *Dorea*, *Ruminococcaceae*_*UBA1819*, *Oscillospiraceae_UCG003*, and *Eubacterium ventriosum* group, correlated with a less than average improvement, and three, *Ruminococcaceae*_*DTU089*, *Prevotella*, and *Adlercreutzia*, with a greater than average improvement in anxiety scores (Table 1). An increasing abundance of genes of the bacterial functional pathway kynurenine synthesis (MGB004) was associated with a less than average improvement in anxiety symptoms at follow-up (Table 1). Appendix A provide summary statistics and boxplots of the genetic scores and microbiome factors significantly associated with improvement of anxiety between T1 and T0.

After the dietary and lifestyle intervention, 51% of the 135 individuals studied reported improving their depression symptoms, with 21%, 21%, and 9% reporting improvement in 1, 2, and 3 scale points, respectively, and 49% reporting no improvement. Two genetic scores were directly associated with a greater than average decrease in intensity scores, OSA and AUD, and height was associated with a less than average improvement (Table 2). Nine bacterial genera and four functional pathways were associated with improvement in depression intensity, eight of which were associated with less than average improvement and five with greater than average improvement (Table 2). Appendix A provide summary statistics and boxplots of the genetic scores and microbiome factors significantly associated with improvement of depression between T1 and T0.

After the dietary and lifestyle intervention, 66% of the 154 individuals studied reported improving their insomnia symptoms, with 29%, 18%, and 19% reporting improvement in 1, 2, and 3 scale points, and 34% reporting no improvement. Two genetic scores were statistically significant, with one associated with a greater than average decrease in intensity scores, type 1 diabetes (T1D), and one associated with a less than average decrease in intensity scores, type 2 diabetes (T2D) (Table 3). Two bacterial genera were associated with improvement in insomnia intensity. *Butyricimonas* was associated with a less than average improvement, and *Roseburia* with a greater than average improvement (Table 3). Finally, one functional pathway was associated with a greater than average improvement in insomnia, nitric oxide synthesis II (nitrite reductase) (Table 3). Appendix A provide summary statistics and boxplots of the genetic scores and microbiome factors significantly associated with the improvement of insomnia between T1 and T0.

### 3.4. At Baseline, Psychiatric Disorders’ Genetic Scores Are Associated with Anxiety or Depression, Whereas Microbial Metabolic Pathways Associate with Sleep Problems

At baseline (T0), individuals that reported depression or anxiety, compared with those who did not report it, had higher mean BMI (*t*-test *p*-value = 0.022), higher proportion of females $\chi_{1}^{2}$ *p*-value = 0.021), lower mean age (*t*-test *p*-value = 2.2 × 10^−5^), and higher prevalence of FGIDs ( $\chi_{1}^{2}$ *p*-value = 8 × 10^−5^). Individuals who reported sleep problems at baseline compared with those who did not report it, had a higher prevalence of FGIDs ( $\chi_{1}^{2}$ *p*-value = 0.021) (Appendix A). We also noted significant differences in overall bacterial diversity between individuals with and without anxiety or depression at the baseline (PERMANOVA, *p* = 0.035; Appendix A).

Two genetic scores, namely alcohol use disorder (AUD) and major depressive disorder (MDD), were significantly associated with an increased prevalence of depression or anxiety (Table 4). Two microbial functional pathways were associated with sleep problems at baseline: (a) menaquinone synthesis (vitamin K2) I (MGB040) with increased prevalence of sleep problems at T0, and (b) inositol degradation (MGB038) with decreased prevalence of sleep problems at T0 (Table 5). Appendix A provide summaries of all variables and their statistical associations with anxiety or depression at baseline, and sleep problems at baseline, respectively. Appendix A provides a summary of the statistics, and Appendix A provide boxplots of the genetic scores associated with anxiety or depression at baseline, and for the microbial functions associated with sleep problems at baseline, respectively.

### 3.5. Multi-Omics Models Are Better Correlated with Mental Health Improvement Than Demographics Models Alone

We compared the ability of models combining demographic (D), genetic (G), and microbiome (M) information to explain the study outcomes: anxiety or depression at baseline, sleep problems at baseline, and improvement of anxiety, depression, and insomnia from T0 to T1. Except for the improvement of insomnia symptoms, we found a trend for D + M or D + G models to explain more of the variation of the outcomes than the D model. Additionally, the D + M + G models were always better than the D models independently or were at least of a similar magnitude as the best D + M or D + G model (Appendix A).

### 3.6. Medication and Recreational Drug Use Do Not Confound Microbiome Associations with Mental Health

We performed sensitivity analyses to evaluate the potential confounding effect of medication, alcohol intake, and recreational drug use on the observed microbiome associations. First, we performed a PERMANOVA, testing the impact of medicines at T0 on all bacterial genera and gut–brain modules/functions (Appendix A). Although we observed a marginally significant effect of anxiolytic medications (*p*-value = 0.04), we did not find any evidence of a confounding effect of anxiolytic or antidepressant drugs on either anxiety or depression at T0 (*p*-value = 0.517 and 0.762, respectively) or sleep problems at T0 (*p*-value = 0.82 and 0.62, respectively). Similarly, we did not find any confounding effect of medications when the analysis was repeated with the subset of microbial markers (both bacterial genera and functions) that were found to be significantly associated with the mental health status at baseline and change in intensity of a particular outcome at T1 (Appendix A). Lastly, we performed the same analyses for each genus and pathway separately. We found no confounding effect of medication, alcohol intake, or recreational drug use on the bacterial genera and pathways identified as significantly associated with the outcomes. The only marginal association, which did not pass multiple testing corrections, was MGB038 (inositol degradation pathway) and alcohol consumption associated with sleep problems at T0 (*p*-value = 0.017). We also noted that demographic factors such as age, gender, and BMI explained most of the variation in the gut microbiome at baseline. We accounted for these factors as covariates in all the statistical models, including the multivariate models. We observed no significant confounding effect of these factors on the associations of the microbiome with mental health outcomes (Appendix A). Furthermore, we also did not find any significant association between alcohol consumption or use of recreational drugs and baseline gut microbiome (Appendix A).

## 4. Discussion

We identified demographic, genetic, and gut microbiome factors that correlate with the mental health status at baseline and future improvement in mental health outcomes, particularly depression, anxiety, and insomnia. Overall, study participants lost on average 5.4% body weight during the study, and more than 95% reported having an improvement in at least one mental health outcome (Appendix A).

This study identified 8 genetic scores, 15 microbiome genera, and 7 functional pathways associated with improvement in anxiety, depression, insomnia, or anxiety/depression and sleep problems at baseline. We noted an association between a higher abundance of kynurenine synthesis (MGB004) and a less than average improvement in anxiety intensity. Kynurenine is a catabolic product of the tryptophan-kynurenine metabolism, and it is further metabolized into kynurenic acid or quinolinic acid. Previous studies have reported higher levels of plasma kynurenine to be associated with anxiety [49] and higher plasma levels of quinolinic acid to be associated with depression [50,51]. The gut microbial pathway involved in p-cresol synthesis (MGB015) was strongly associated with a less than average improvement in depression intensity. Although previous reports have linked this gut microbial metabolite with autism [52], its role in depression has not been previously reported.

Interestingly, nitric oxide synthesis II (MGB026: nitrite reductase) was significantly associated with a greater than average improvement in depression and insomnia intensity. In contrast, the nitric oxide degradation I pathway (MGB027: nitric oxide dioxygenase) was associated with a greater than average improvement in depression intensity only. Increased NO levels have been found in the plasma of MDD patients (see [53]). Antidepressants and anxiolytics have been shown to induce inhibition of NO synthesis (see [54] and references therein). Therefore, increased NO degradation by the gut microbiome may mimic the effects of pharmacological treatments. Contrastingly, the research literature also suggests that higher NO levels may benefit mental health due to their role in neuronal plasticity, inflammation, and oxidative stress [55]. Given its intrinsic properties enabling rapid diffusion and its activation of signaling cascades with functions in multiple physiological contexts, it is not surprising that the mechanistic link between NO degradation or synthesis by the gut microbiome and human behaviors is not well understood yet [54,56].

The association between the increasing abundance of butyrate synthesis II (MGB053) and a less than average improvement in depression is in the opposite direction of the generally reported relationship between short-chain fatty acids (SCFAs) and mental health [54,57]. These previous reports linked lower butyrate with poorer mental health before intervention. However, the association found in this study relates to the effect of the intervention. Thus, it reflects that a high abundance of butyrate synthesis genes at baseline is associated with improvement, but less so than the average. This could be explained because the intervention increases dietary fiber, which is known to increase the relative abundance of butyrate-producing microbes and those producing other SCFAs. Therefore, having a high baseline abundance of butyrate synthesis genes may limit the beneficial effect an individual can attain during the intervention. We also noted that the literature reports that the beneficial or detrimental effects of higher butyrate synthesis by the gut microbiome may depend on the context, such as the section of the intestine where the butyrate-producing microbes are inhabiting [58].

Previous research has pointed to the association between depression and *Oscillospiraceae_UCG003* [59], *Eubacterium ventriosum* group [60,61], *Lactobacillus* [62], *Prevotella* [63], and anxiety with *Ruminococcaceae*_*UBA1819* [64] and *Ruminococcaceae*_*DTU089* [65] and the direction of the association was concordant with previous reports. *Butyricimonas* and *Roseburia* were associated with improvement in insomnia, replicating previously reported associations [66]. Interestingly, in our dataset, several genera were systematically associated with anxiety or depression at baseline and with improvement in multiple outcomes. For instance, the *Eubacterium ventriosum* group was significantly associated with improvement in anxiety and depression with the same direction of effect (beta = 0.24 and 0.20). Likewise, *Prevotella* was associated with improvement in anxiety and depression with the same direction of effect (beta = −0.13 and −0.12) and nominally associated with anxiety or depression at baseline (beta = −0.3 and *p*-value = 0.0028) (Appendix A). *Oscillospiraceae_UCG003* was associated with improvement on depression (beta = 0.28) and reached a nominal association with improvement on anxiety (beta = 0.42 and *p*-value = 0.0048) (Appendix A). In fact, there were no genera associated (with *p*-values < 0.05) with multiple outcomes with an inconsistent direction of effect.

The association between the IBS genetic score and a less than average improvement in anxiety suggests that individuals at higher inherited risk for IBS improve their anxiety symptoms less than those without the risk after our digital therapeutics intervention. Cameron et al. [25] had already reported a positive correlation between genetic risk for IBS and anxiety. Furthermore, Eijsbouts et al. (2021) suggested a common basis for anxiety and IBS independent of their comorbidity [13]. Our results suggest that this shared etiology may also have implications for therapeutic response. The association between BMI and OSA genetic scores with a greater than average improvement in anxiety is likely due to the direct relationship between obesity and the occurrence of OSA [62,67]. In fact, OSA partially causes sleep disturbances and poor sleep and is also associated with higher anxiety symptoms [68,69]. Therefore, it is plausible that weight loss would lead to a reduction in OSA and anxiety. In line with this hypothesis, examining the ratio of the linear regression coefficients for improvement of anxiety for different weight loss groups shows that subjects who did not lose or gain body weight (no change group) improved 17% less (no change/3–5% weight loss = 0.6163/−3.6632 = −0.168, *p*-value = 0.003), and those that gained weight (weight gain group) improved 62% less (weight gain/3–5% weight loss = 2.273/−3.6632 = −0.620, *p*-value = 0.025) than those that lost between 3 and 5 percent body weight (Appendix A). Other weight loss groups did not differ significantly in improvement compared to the 3–5% weight loss group.

Improvement in depression was associated with the AUD genetic score, with a higher genetic score implying a more significant improvement in self-reported depression. Genetic scores for alcohol use are positively correlated with the amount of alcohol consumed. Avoiding alcohol consumption is strongly recommended as part of the digital therapeutics intervention and removing alcohol from the diet would lead to improvement in mental health.

Epidemiological studies have shown significant comorbidity between insomnia and type 1 and type 2 diabetes [70,71], and there exists evidence showing that weight loss is associated with improvement in sleep and insomnia [72]. In line with our results associating the T1D genetic score with a greater than average improvement of insomnia, previous research has linked autoimmune disease and T1D in particular with increased risk of insomnia. Interestingly, we also found the T2D genetic score associated with a less than average improvement in insomnia. In addition to their co-occurrence, there is evidence of a genetic [73] and a causal link between T2D and insomnia [74]. The metabolic nature of T2D, compared with T1D, is more prone to improvement under the implemented dietary intervention, which is focused, among other objectives, on reducing insulin resistance and diabetes severity and risk, which could explain why subjects with higher genetic risk for T2D may improve less than average on their insomnia.

Subjects’ reported anxiety or depression at baseline was associated with the genetic scores of two psychiatric disorders, namely alcohol use disorder (AUD) and major depressive disorder (MDD). These two associations align well with known genetic correlations of AUD and MDD with mood and anxiety disorders [75,76]. We also identified an association between gut microbial menaquinone synthesis (vitamin K2) I (MGB040) and a higher prevalence of sleep problems at baseline. There is a paucity of evidence documenting this relationship, but previous reports align with our findings [77,78,79,80] and warrant additional investigation due to their potential to support nutrigenomics interventions. Similarly, there was a significant association between microbial inositol degradation (MGB038) and a lower prevalence of sleep-related issues at baseline. Although there exists no direct evidence of the relationship between gut microbial-based inositol degradation and sleep improvement, earlier reports suggested an association of frontal cortex myo-inositol concentration with sleep and depression [81].

Our study evaluated the association between genetic scores and microbiome factors with the improvement in depression, anxiety, and insomnia after a dietary and lifestyle intervention. Therefore, this study assessed a gene and microbiome by environment interaction where the individuals responded and improved differentially depending on their genetic and baseline microbiome profile. Previous research has focused on identifying genetic and microbiome correlations and signatures with disease diagnosis or symptom severity (for instance, see [5,16,82,83]). Currently, it is unknown whether diagnostic associations with genetic or microbiome factors are also associated with prognosis after an intervention, especially since physiological mechanisms might differ based on the interventions. To approach this question, we compared the association between genetic and microbiome factors with baseline (T0) and follow-up (T1) responses by participants. We found the effect sizes of the diagnostic and prognostic associations were correlated but weakly so, with microbial functional pathways having the largest r-squared values, for instance, 0.38 for the associations between anxiety or depression at T0 with depression improvement at T1 (Appendix A). Several of the associations with improvement at T1 had been previously reported as diagnostic associations suggesting that, to some extent, the same physiological process may be associated with the development of the conditions and their improvement.

This study has some limitations that are important to note. Firstly, the findings from this study are derived from a weight loss cohort and thus may be only reflective of the population with mental health that is overweight or obese. Secondly, this study did not consider factors known to influence the microbiome composition, such as diet, other disease diagnoses, measures of environmental health, social determinants of health, or any other situational factors that may confound the results presented. Our statistical models accounted for potential confounders and the evidence presented indicates that these do not affect the interpretation of our findings (Appendix A). However, we cannot rule out their effect due to the intrinsic limitations of our study design. Thirdly, the survey instrument utilized was not a clinically validated questionnaire. Additionally, the questionnaire was applied at follow-up (T1) retrospectively for the baseline (or T0) time point so that the data may contain recall bias. However, evidence shows that non-clinical and self-reported assessment of previous events of depression is a valid construct [75,83]. In addition, we observed a high concordance, 81% (74 out of 91), between the self-reported measure of depression or anxiety and the self-reported use of medication for either condition, in line with published research [84]. Fourthly, our associations with bacterial functional pathways were based on predicting the abundance of the relevant genes and do not directly relate to the enzymatic or molecular functions at strain-specific levels.

## 5. Conclusions

Overall, the evidence gathered in this study supports the notion that weight loss is associated with improvement in mental health and that genetic and gut microbiome factors explain the heterogeneity in the level of improvement among subjects. Genetic and gut microbiome factors contribute to and mediate individuals’ improvement in mental health after dietary intervention with relative importance equal to or greater than that of demographic characteristics on their own. Our results provide evidence of the value of genetic and microbiome factors to explain future mental health improvement and guide personalized interventions. We believe our findings are relevant to the large population of individuals affected by mental health issues and obesity, and we found that they are also congruent with previous reports. However, they warrant validation via replication on independent samples and follow-up studies using metabolomics, longitudinal microbiome sampling, and other assays to understand the associations more deeply.

## Figures and Tables

**Table 1 jpm-12-01237-t001:** Variables associated with improvement of anxiety between T1 and T0. N samples correspond to the number of samples included in the analyses. In the case of microbiome analysis, it corresponds to the number of samples with an abundance greater than zero. IBS = irritable bowel syndrome, BMI = body mass index, OSA = obstructive sleep apnea, and FDR = false discovery rate. Beta and ‘beta se’ correspond to the regression coefficient and its standard error, respectively. *p*-value of statistical test evaluating beta ≠ 0.

Variable	N Samples	Beta *	Beta Se	*p*-Value	FDR
Genetics					
IBS	135	0.248	0.0788	0.0016	0.015
BMI	135	−0.218	0.0704	0.0018	0.015
OSA	135	−0.232	0.0869	0.0075	0.04
Microbial Taxa					
*Dorea*	118	0.250	0.081	0.0017	0.071
*Ruminococcaceae UBA1819*	76	0.305	0.102	0.0028	0.071
*Ruminococcaceae DTU089*	64	−0.385	0.133	0.0039	0.071
*Prevotella*	51	−0.128	0.045	0.0043	0.071
*Oscillospiraceae UCG003*	65	0.418	0.148	0.0048	0.071
*Eubacterium ventriosum group*	97	0.238	0.0931	0.011	0.13
*Adlercreutzia*	77	−0.469	0.190	0.014	0.15
Microbial Functions					
MGB004: Kynurenine synthesis	135	0.383	0.131	0.0033	0.14

***** A value of beta > 0 indicates increasing values of the microbiome or genetic variable are associated with less than average improvement and a value of beta < 0 that the improvement is greater than average.

**Table 2 jpm-12-01237-t002:** Variables associated with improvement of depression between T1 and T0. N samples correspond to the number of samples included in the analyses. In the case of microbiome analysis, it corresponds to the number of samples with an abundance greater than zero. OSA = obstructive sleep apnea, AUD = alcohol use disorder, and FDR = false discovery rate. ‘Beta’ and ‘beta se’ correspond to the regression coefficient and its standard error, respectively. *p*-value of statistical test evaluating beta ≠ 0.

Variable	N Samples	Beta *	Beta Se	*p*-Value	FDR
Genetics					
OSA	135	−0.235	0.088	0.0075	0.12
AUD	135	−0.218	0.096	0.023	0.14
Height	135	0.168	0.076	0.027	0.14
Microbial Taxa					
*Clostridium innocuum* group	52	0.183	0.060	0.0022	0.132
*Oscillospiraceae UCG003*	68	0.283	0.0991	0.0043	0.132
*Anaerostipes*	132	0.202	0.0751	0.0071	0.132
*Eubacterium ventriosum* group	98	0.196	0.0746	0.0086	0.132
*Lactobacillus*	70	−0.189	0.0751	0.011	0.132
*Negativibacillus*	71	0.279	0.114	0.014	0.132
*Prevotella*	52	−0.116	0.0475	0.015	0.132
*Oscillibacter*	125	0.178	0.0732	0.015	0.132
*Actinomyces*	70	−0.508	0.211	0.016	0.132
Microbial Functions					
MGB053: Butyrate synthesis II	135	0.849	0.260	0.0011	0.043
MGB015: p-Cresol synthesis	135	0.972	0.341	0.0044	0.078
MGB026: Nitric oxide synthesis II (nitrite reductase)	92	−0.167	0.0607	0.0058	0.078
MGB027: Nitric oxide degradation I (NO dioxygenase)	135	−0.171	0.0666	0.0099	0.098

***** A value of beta > 0 indicates increasing values of the microbiome or genetic variable are associated with less than average improvement and a value of beta < 0 that the improvement is greater than average.

**Table 3 jpm-12-01237-t003:** Variables associated with improvement of insomnia between T1 and T0. N samples correspond to the number of samples included in the analyses. In the case of microbiome analysis, it corresponds to the number of samples with an abundance greater than zero. T1D = type 1 diabetes, T2D = type 2 diabetes, and FDR = false discovery rate. Beta and ‘beta se’ correspond to the regression coefficient and its standard error, respectively. *p*-value of statistical test evaluating beta ≠ 0.

Variable	N Samples	Beta *	Beta Se	*p*-Value	FDR
Genetics					
T2D	154	0.125	0.050	0.013	0.15
T1D	154	−0.148	0.063	0.018	0.15
Microbial Taxa					
*Butyricimonas*	68	0.319	0.086	2.2 × 10^−4^	0.018
*Roseburia*	144	−0.143	0.043	7.5 × 10^−4^	0.0291
Microbial Functions					
MGB026: Nitric oxide synthesis II (nitrite reductase)	108	−0.185	0.0418	9 × 10^−6^	3.8 × 10^−4^

***** A value of beta > 0 indicates increasing values of the microbiome or genetic variable are associated with less than average improvement and a value of beta < 0 that the improvement is greater than average.

**Table 4 jpm-12-01237-t004:** Variables associated with depression or anxiety at T0. Summary of statistical associations with depression or anxiety at baseline. N samples correspond to the number of samples included in the analyses. AUD = alcohol use disorder, MDD = major depressive disorder, and FDR = false discovery rate. Beta and ‘beta se’ correspond to the regression coefficient and its standard error, respectively. *p*-value of statistical test evaluating beta ≠ 0.

Variable	N Samples	Beta *	Beta Se	*p*-Value	FDR
Genetics					
AUD	328	0.361	0.139	0.0096	0.13
MDD	328	0.309	0.129	0.016	0.13

* A value of beta > 0 indicates increasing values of the microbiome or genetic variable are associated with an increasing prevalence of anxiety or depression and a value of beta < 0 indicates a lower prevalence of anxiety or depression.

**Table 5 jpm-12-01237-t005:** Variables associated with sleep problems at T0. Summary of statistical associations with sleep problems at baseline. N samples correspond to the number of samples included in the analyses. In the case of microbiome analysis, it corresponds to the number of samples with an abundance greater than zero. FDR = false discovery rate. Beta and ‘beta se’ correspond to the regression coefficient and its standard error, respectively. *p*-value of statistical test evaluating beta ≠ 0.

Variable	N Samples	Beta *	Beta Se	*p*-Value	FDR
Microbial Functions					
MGB040: Menaquinone synthesis (vitamin K2) I	328	0.863	0.298	0.0038	0.092
MGB038: Inositol degradation	328	−0.484	0.170	0.0044	0.092

* A value of beta > 0 indicates increasing values of the microbiome or genetic variable are associated with an increasing prevalence of sleep problems and a value of beta < 0 indicates a lower prevalence of sleep problems.

## Data Availability

The microbiome sequence data used in this study were submitted to NCBI SRA under Bioproject accession number PRJNA821674.

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
