# Peer review of "Mental Health Symptom Reduction Using Digital Therapeutics Care Informed by Genomic SNPs and Gut Microbiome Signatures"

_jpm, 2022, doi:10.3390/jpm12081237_

Round 1

Reviewer 1 Report

In this manuscript, the authors evaluated the association between genetic scores and microbiome factors with the improvement in depression, anxiety, and insomnia after a dietary and lifestyle intervention by comparing the associations between genetic and microbiome factors with baseline (T0) and follow-up (T1) responses by participants in Digbi Health’s personalized digital therapeutics care program. They identified a few genetic scores, microbiome genera, and functional pathways that showed association with improvement in health status observed. The data generated from current study provide clues how genetic and gut microbiome factors affect weight loss associated mental health improvement. Overall, the manuscript is well written, well organized and comprehensively described. Comments regarding the manuscript below.

1.     As the authors stated in discussion, one limitation on bacterial functional pathway analysis was based on prediction using 16S rRNA gene sequencing. How likely would your prediction reflect true bacterial functional pathways? Can you perform WGS on some samples to validate your prediction?

2.     In your 16S rRNA sequencing analysis, rarefaction was used and yielded a depth of 36k reads per sample. Using this cutoff, how many sample were excluded and how many (proportion) reads were excluded? And whether the choice of this cutoff is reasonable?

3.     In your data analysis, e.g., logistic/linear regression models, can you provide github links for your script?

4.     In the analysis, the small sample size of individuals with worsening symptoms were excluded from additional analyses. Did you have a chance to look at these data and whether they showed a potential opposite associations?

Author Response

Reviewer: In this manuscript, the authors evaluated the association between genetic scores and microbiome factors with the improvement in depression, anxiety, and insomnia after a dietary and lifestyle intervention by comparing the associations between genetic and microbiome factors with baseline (T0) and follow-up (T1) responses by participants in Digbi Health’s personalized digital therapeutics care program. They identified a few genetic scores, microbiome genera, and functional pathways that showed association with improvement in health status observed. The data generated from current study provide clues how genetic and gut microbiome factors affect weight loss associated mental health improvement. Overall, the manuscript is well written, well organized and comprehensively described. Comments regarding the manuscript below.

Response: Thank you for your thoughtful review and constructive comments.

Reviewer: The authors stated in discussion, one limitation on bacterial functional pathway analysis was based on prediction using 16S rRNA gene sequencing. How likely would your prediction reflect true bacterial functional pathways? Can you perform WGS on some samples to validate your prediction?

Response: Thank you for your comment. As mentioned in the manuscript, we acknowledge the fact that the bacterial functional pathways were based on predicting the abundance of the relevant genes and are not an exact reflection of the gene abundances for enzymatic activities at strain level. We have now added more clarity on this limitation in the manuscript.

Regarding the prediction accuracy of the approach: The pathway abundances were inferred using the software PICRUSt2 (Douglas et al., 2020) on the basis of ancestral gene content reconstruction, which is a very conservative approach. In the manuscripts describing their software, the authors of PICRUSt have provided evidence that the predicted functions are highly correlated with shotgun metagenome data, specially for human microbiome samples. For instance, for genetic information processing, the authors showed a mean accuracy of prediction of 0.99 ± 0.03 s.d., and for environmental information processing a mean accuracy of 0.95 ± 0.04 s.d.

Reviewer: In your 16S rRNA sequencing analysis, rarefaction was used and yielded a depth of 36k reads per sample. Using this cutoff, how many sample were excluded and how many (proportion) reads were excluded? And whether the choice of this cutoff is reasonable?

Response: Following are our point-wise responses to the queries on rarefaction:

A - The rarefaction depth was decided as 36,000 reads per sample based on the minimum number  of reads per sample across the 344 samples in the study (n=36,365). There was no further loss/exclusion of samples after rarefying the data at this depth, so no samples were excluded from analysis.

B - By rarefying samples to 36,000 reads, we retained around 30% of reads per sample. In order to best preserve the diversity of the full sample we performed 500 iterations of rarefaction, which gives us high confidence of having sampled enough subsets. The number of rarefaction iterations  was not mentioned in the original manuscript, so have added it explicitly.  

C - Considering the variation in the library sizes, and the fact that we didn’t want to discard samples from the study, rarefaction to the minimum number of reads in a sample, with 500 iterations, seemed as a reasonable solution. Hope this provides more clarity to the reviewer about our decision making process for rarefaction.  

Reviewer: In your data analysis, e.g., logistic/linear regression models, can you provide github links for your script?

Response: Thank you for your comment. We have added a section to the Supp Methods section of the article under the title “Example code for the regression statistical analyses” and we have made the code also available as an ipython notebook https://gist.github.com/inti4digbi/a7de296912342c7a6e46505f51fc2875. These include: a) sample data which was generated by randomizing real data used on the study,  and b) example Python code using the same options used for the analyses performed on the study. We believe this will allow others to reproduce the analyses if using the same data or perform the same or similar analyses in other datasets.

Reviewer: In the analysis, the small sample size of individuals with worsening symptoms were excluded from additional analyses. Did you have a chance to look at these data and whether they showed a potential opposite associations?

Response: Thank you for your comment. As mentioned in the manuscript, most individuals (>95%) reported improvement or maintenance of symptoms. Only 4% (6 out of 148) for depression, 2.7% (4 out of 147) for anxiety, and 1.8% (3 out of 163) for insomnia reported worsening of symptoms (Figure S2 and Table S4). Due to the small sample size of individuals with worsening symptoms, we excluded them from additional analyses. Considering that the sample size was too small to make any meaningful interpretations, especially for the microbiome data (where the inter-individual variation is usually high), we did not analyze them further. However, we acknowledge that this is an important point and would certainly incorporate it in our future research studies with a larger and more balanced sample size.

End of reply to reviewer comments and questions.

Reviewer 2 Report

Overall, the manuscript is interesting. It may be interesting to those who are working in this field. However, this manuscript is with too many typos and grammar problems. I suggest that the author ask a native English speaker to edit your manuscript. Here, I included some corrections here:

Abstract: Actually, i can't get the meaningful information in the abstract section. 1. The description of the purpose, method and results of current experiment is not clear. 2. It is too cumbersome to confuse readers, even cannot find the core significance of this test.

How to evaluate the improvement in mental health? How to connect the mental health and weight loss? Is there any standard?

Many objective factors affect the gut microbiota composition of animals, such as age and sex. How to exclude the influence of objective factors on the gut microbiota?

“Our results provide evidence of the value of genetic and microbiome factors to explain future mental health improvement and guide personalized interventions.” I can't find any evaluation criterion about genetic and microbiome in current research. And, How to control the experimental single variable? genetic? or microbiome factor?

Author Response

Reviewer: Overall, the manuscript is interesting. It may be interesting to those who are working in this field. However, this manuscript is with too many typos and grammar problems. I suggest that the author ask a native English speaker to edit your manuscript. Here, I included some corrections here:

Reviewer: Abstract: Actually, i can’t get the meaningful information in the abstract section. 1. The description of the purpose, method and results of current experiment is not clear. 2. It is too cumbersome to confuse readers, even cannot find the core significance of this test.

Response: We thank the reviewer for his/her comment and we have worked to improve the clarity and readability of the document. We have requested a native English speaker colleague, who is also a University Professor of Writing & Communication, to review the document. As a result, we have made multiple changes throughout the document to improve grammar, typos, flow, and readability. We are confident this new version is clearer and more appropriate for the readership of the journal.

Reviewer: How to evaluate the improvement in mental health? How to connect the mental health and weight loss? Is there any standard?

Response: Thank you, we agree with the reviewer's comment. We acknowledge there may be lack of clarity on how the improvement in mental health was measured and we have made clearer on the methodology how the mental health improvement was measured. We used an online questionnaire that was provided to the study participants when they reached 2% body weight loss or more and at that point they rated their baseline (start of the intervention) and current (at the moment of answering) status for anxiety, depression, and insomnia using a 0 to 5 (from good to bad health) severity scale. The link with body weight is established using the amount of weight loss during the time of the intervention. We used the regression models to statistically link the two as reported in Tables S6 to S8. In the discussion, we present the limitations of the study and discuss the instrument used to measure mental health improvement and its implications for the conclusions presented in the article. 

Reviewer: Many objective factors affect the gut microbiota composition of animals, such as age and sex. How to exclude the influence of objective factors on the gut microbiota?

Response: Thank you, we agree with the reviewer's comment. All statistical models used included as covariates FGID: binary, the self-reported status of a functional gastrointestinal disorder; gender: binary, male or female; BMI at T0: continuous variable; Age: continuous variable; weight loss: categorical variable, categorized as those with no change, lost 0 to 5%, 5-10% or more than 10% of their body weight at T1 in relation to T0; and five principal components (continuous variable) calculated using the genetic ancestry analyses described in the manuscript. Therefore, the results presented on the main text, including Tables 1 to 5, have already accounted for the potential confounding effect of these variables. 

Additionally, we performed statistical analyses with the objective of testing the confounding effect of medication intake (anxiolytics or antidepressants), gender, alcohol consumption, and use of recreational drugs on the microbiome. Table S13 provides the results of these analyses that were performed at three levels of granularity: on the whole microbiome profile using beta diversity and permanova tests (Table S13-A, S13-B, S13-C, and S13-D), on the group of statistically significant microbial markers (both bacterial genera and functional pathways) using beta diversity and permanova tests (Table S13-E and S13-F), and individually on each genus and functional pathway using linear regression models (Table S13-G). The results from these analyses did not provide statistical evidence of the confounding effect and association of these factors on the interindividual differences in the microbiome.

We cannot completely rule out the effect of these confounding factors due to the limitations of our study design, but we believe the evidence collected on this study suggests these factors are not a significant confounder of the results we have presented. We have added a sentence to the discussion as part of the section describing the limitations of the study raising the issues you have brought our attention to.

Reviewer: “Our results provide evidence of the value of genetic and microbiome factors to explain future mental health improvement and guide personalized interventions.” I can’t find any evaluation criterion about genetic and microbiome in current research. And, How to control the experimental single variable? genetic? or microbiome factor?

Response: We thank the reviewer's comment. We performed statistical analyses to measure the relative ability or the correlation between genetic or microbiome factors with improvement in mental health. This can be interpreted as a measure of how relevant each of the two datasets are in regards to explaining improvement in the sense that if one has a higher correlation than the other it is deemed as more important. For these analyses, we used a cross-validation analysis and the r-squared as the criteria to measure the fit of the model on independent samples (albeit from the same cohort studied). The relevance of genetic or microbiome factors was deemed on the basis of comparing the r-squared values. However, we agree with the reviewer that since the individuals have all been “exposed” to the genetic and microbiome information by means of their nutritional reports and coaching, it is not really possible to separate the effect of each component (genetic or microbiome) on the success of the intervention. We believe it is valid and interesting to quantify which of the two data types seems more relevant as therapeutic biomarkers. We have changed passages in the discussion to make this point clearer to the reader.

End of reply to reviewers comments and questions.

Round 2

Reviewer 2 Report

None.